# Anti-*Salmonella* Defence and Intestinal Homeostatic Maintenance In Vitro of a Consortium Containing *Limosilactobacillus fermentum* 3872 and *Ligilactobacillus salivariu*s 7247 Strains in Human, Porcine, and Chicken Enterocytes

**DOI:** 10.3390/antibiotics13010030

**Published:** 2023-12-28

**Authors:** Vyacheslav M. Abramov, Igor V. Kosarev, Andrey V. Machulin, Evgenia I. Deryusheva, Tatiana V. Priputnevich, Alexander N. Panin, Irina O. Chikileva, Tatiana N. Abashina, Ashot M. Manoyan, Anna A. Akhmetzyanova, Dmitriy A. Blumenkrants, Olga E. Ivanova, Tigran T. Papazyan, Ilia N. Nikonov, Nataliya E. Suzina, Vyacheslav G. Melnikov, Valentin S. Khlebnikov, Vadim K. Sakulin, Vladimir A. Samoilenko, Alexey B. Gordeev, Gennady T. Sukhikh, Vladimir N. Uversky, Andrey V. Karlyshev

**Affiliations:** 1Federal Service for Veterinary and Phytosanitary Surveillance (Rosselkhoznadzor) Federal State Budgetary Institution “The Russian State Center for Animal Feed and Drug Standardization and Quality” (FGBU VGNKI), 123022 Moscow, Russia; 2Kulakov National Medical Research Center for Obstetrics, Gynecology and Perinatology, Ministry of Health, 117997 Moscow, Russiaa_gordeev@oparina4.ru (A.B.G.);; 3Skryabin Institute of Biochemistry and Physiology of Microorganisms, Federal Research Center “Pushchino Scientific Center for Biological Research of Russian Academy of Science”, Russian Academy of Science, 142290 Pushchino, Russia; 4Institute for Biological Instrumentation, Federal Research Center “Pushchino Scientific Center for Biological Research of Russian Academy of Science”, Russian Academy of Science, 142290 Pushchino, Russia; 5Blokhin National Research Center of Oncology, Ministry of Health RF, 115478 Moscow, Russia; 6Alltech Company Moscow, 105062 Moscow, Russia; 7Federal State Educational Institution of Higher Professional Education, Moscow State Academy of Veterinary Medicine and Biotechnology Named after K.I. Skryabin, 109472 Moscow, Russia; ilnikonov@yandex.ru; 8Gabrichevsky Research Institute for Epidemiology and Microbiology, 125212 Moscow, Russia; 9Institute of Immunological Engineering, 142380 Lyubuchany, Russia; vkhleb@mail.ru (V.S.K.); svk@bio-chehov.ru (V.K.S.); 10Department of Molecular Medicine, Morsani College of Medicine, University of South Florida, Tampa, FL 33612, USA; vuversky@usf.edu; 11Department of Biomolecular Sciences, School of Life Sciences, Chemistry and Pharmacy, Faculty of Health, Science, Social Care and Education, Kingston University London, Kingston upon Thames KT1 2EE, UK

**Keywords:** *Salmonella* enteritidis, multi-drug resistance, gene expression

## Abstract

*Limosilactobacillus fermentum* strain 3872 (LF3872) was originally isolated from the breast milk of a healthy woman during lactation and the breastfeeding of a child. *Ligilactobacillus salivarius* strain 7247 (LS7247) was isolated at the same time from the intestines and reproductive system of a healthy woman. The genomes of these strains contain genes responsible for the production of peptidoglycan-degrading enzymes and factors that increase the permeability of the outer membrane of Gram-negative pathogens. In this work, the anti-*Salmonella* and intestinal homeostatic features of the LF3872 and LS7247 consortium were studied. A multi-drug resistant (MDR) strain of *Salmonella* enteritidis (SE) was used in the experiments. The consortium effectively inhibited the adhesion of SE to intact and activated human, porcine, and chicken enterocytes and reduced invasion. The consortium had a bactericidal effect on SE in 6 h of co-culturing. A gene expression analysis of SE showed that the cell-free supernatant (CFS) of the consortium inhibited the expression of virulence genes critical for the colonization of human and animal enterocytes. The CFS stimulated the production of an intestinal homeostatic factor—intestinal alkaline phosphatase (IAP)—in Caco-2 and HT-29 enterocytes. The consortium decreased the production of pro-inflammatory cytokines IL-8, TNF-α, and IL-1β, and TLR4 mRNA expression in human and animal enterocytes. It stimulated the expression of TLR9 in human and porcine enterocytes and stimulated the expression of TLR21 in chicken enterocytes. The consortium also protected the intestinal barrier functions through the increase of transepithelial electrical resistance (TEER) and the inhibition of paracellular permeability in the monolayers of human and animal enterocytes. The results obtained suggest that a LF3872 and LS7247 consortium can be used as an innovative feed additive to reduce the spread of MDR SE among the population and farm animals.

## 1. Introduction

*Salmonella*, a type of bacteria, is a facultative anaerobic intracellular pathogen classified as Gram-negative and as part of the Enterobacteriaceae family [1]. The dominant nontyphoidal serovars of *Salmonella enterica*, *S.* enteritidis (SE), and *S.* typhimurium (ST), belong to common foodborne pathogens and are considered socially significant zoo-anthroponotic infections [2,3]. Salmonellosis caused by these two serovars is one of the world’s leading bacterial infectious enteritis [4]. They are responsible for 93 million infections annually [5].

Toward the end of the 20th century, SE became a significant pathogen associated with eggs [6]. Epidemiological information from the European Union, United States, Canada, and various other nations suggests that SE has occupied the ecological niche left vacant after the elimination of the *S*. *enterica* serovar Gallinarum in poultry. This has led to a substantial increase in human infections [7,8,9,10,11,12,13]. SE is responsible for symptoms related to the gastrointestinal system, and certain invasive strains of SE have emerged as a cause of infections beyond the intestinal tract on a global scale [14,15,16,17].

SE infiltrates table eggs through a combination of pathways: horizontal transmission occurs from the feces of infected laying hens, vertical transmission takes place via the yolk or eggshell membranes before egg laying, and contamination of the eggshell occurs after laying [18,19]. Moreover, SE is a common serovar detected in raw poultry products and eggs, and is frequently present in cooked or ready-to-eat dishes that incorporate poultry or eggs as ingredients [20,21]. Salmonellosis of pigs and cattle is also of great economic importance, leading to a decrease in productivity as a result of treatment costs, weight loss, a decrease in meat and milk yields, and animal mortality [22,23,24].

The number of infections caused by multi-drug resistant (MDR) *Salmonella* strains is growing [25]. *Salmonella* MDR strains are carried by migratory birds [26]. SE has been increasingly reported to be resistant to commonly used antimicrobials, limiting therapeutic choices for treating severe forms of infection [27,28,29,30]. SE has become a significant global concern for food safety [31]. *Salmonella* has earned a spot on the World Health Organization’s (WHO) global priority pathogen list of antibiotic-resistant bacteria [32].

Traditionally, antibiotics have been used as a strategy to combat *Salmonella* infections. Nevertheless, the frequent and prolonged utilization of antibiotics not only fosters antibiotic resistance among *Salmonella* serovars but also disrupts the balance of the normal intestinal microbiota [33]. MDR SE strains in farm animals have the potential to directly transmit to humans through the food chain or transfer their resistance genes to human pathogens, facilitated by mobile genetic elements linked to conjugative plasmids [34]. The rapid global spread of MDR SE pathogens poses a significant threat to both humans and animals, emphasizing the need for the development and implementation of alternative methods to antibiotics for addressing these pathogens [35]. *Salmonella* strains resistant to fluoroquinolones, carbapenems, or third-generation cephalosporins have been designated as high-priority pathogens by the WHO [32].

An encouraging alternative approach involves the potential preventive and therapeutic use of probiotics. Previous studies have documented the application of probiotics in preventing and treating gastrointestinal infections caused by *Salmonella* [35,36,37,38,39,40]. *Lactobacillus spp.* were identified as having immunomodulatory and antagonistic properties against *Salmonella* infection, demonstrated both in vitro and in vivo [41,42,43]. The effects of probiotic strains are achieved through immune regulation, particularly by modulating the balance between pro- and anti-inflammatory cytokines (IL-8, TNF-α, and Il-1β) [44]. Toll-like receptors (TLRs) play a crucial role in identifying pathogenic components and organizing responses to specific pathogen-associated molecular patterns. These patterns are associated with a variety of micro-organisms, including bacteria, viruses, and fungi [45,46]. Following infection with intestinal bacteria, the activation of TLR4 may result in the overexpression of inflammatory cytokines through the initiation of their respective signaling pathways [47]. *Lactobacillus* strains have shown the ability to suppress the activation of TLR4 signaling [48,49]. While some probiotic strains have demonstrated increased phagocytosis in phagocytic cells and modified cytokine production following interactions with TLRs in different cell populations, these studies are limited to date, and the results have sometimes been contradictory [50]. Multistrain probiotics are considered more effective than their monostrain counterparts, attributed to the additive and synergistic effects that enhance their overall efficacy [51,52,53]. The anti-*Salmonella* properties of the consortia of probiotic lactobacilli remain poorly studied.

In this work, we prepared a consortium containing two strains of lactobacilli: the *Limosilactobacillus fermentum* strain 3872 (LF3872) strain and *Ligilactobacillus salivarius* strain 7247 (LS7247) strain. LF3872 was originally isolated from the breast milk of a healthy woman during lactation and the breastfeeding of a child (the complete genome sequence of the LF3872 chromosome and plasmid pLF3872 are available at NCB GenBank (https://www.ncbi.nlm.nih.gov/genbank/, accessed on 20 October 2023, accession numbers CP011536.1 and CP011537.1) [54]. Previous studies involving a whole-genome sequence analysis of strain LF3872 allowed the identification of a plasmid carrying genes encoding a putative collagen-binding protein (CBP) and fibronectin-binding protein, as well as other probiotic-related genes [55,56]. The role of CBP in the bacterial attachment to host cells was confirmed after its purification in in vitro binding experiments involving an analog of a host cell receptor (collagen I) [57]. In addition, those experiments demonstrated the ability of LF3872 to compete with cells of a gastrointestinal pathogen *Campylobacter jejuni* for binding to host cell receptors, thus providing a proof of principle that LF3872 may potentially be used for fighting infections caused by this pathogen. LS7247 was simultaneously originally isolated from the intestines and reproductive system of a healthy woman (the complete genome sequence of LS7247(2102-15) chromosome and plasmid are available at NCB GenBank, accession numbers CP090411:CP090413) [58]. The genomes of these strains contain clusters of genes responsible for the production of peptidoglycan-degrading enzymes enterolysin A and metalloendopeptidase, and factors that increase the permeability of the outer membrane (OM) of Gram-negative pathogens [58,59]. To date, the anti-*Salmonella* properties of the LF3872 and LS7247 consortium remain unexplored, as well as the role of the consortium in ensuring intestinal homeostasis.

Thus, the purpose of this work was to evaluate in vitro anti-*Salmonella* and intestinal homeostatic features of a LF3872 and LS7247 consortium in human, porcine, and chicken enterocytes.

## 2. Results

### 2.1. Inhibition of SE Adhesion to Intact Human and Animal Enterocytes by LF3872 and LS7247 Strains and Their Consortium

The SE pathogen demonstrated a high adhesive ability to intact human HT-29 cells (6.3 ± 0.1 log CFU/mL), to intact human Caco-2 cells (6.4 ± 0.1 log CFU/mL), to intact porcine IPEC-J2 cells (5.4 ± 0.1 log CFU/mL), and to intact chicken primary cecal enterocytes (CPCE) cells (5.6 ± 0.1 log CFU/mL) (Figure 1). SE adhesion was significantly (*p* < 0.01) reduced when human and animal enterocytes were pretreated by the LF3872 strain or the LS7247 strain. A combination of strains LF3872 and LS7247 resulted in the increased inhibition of the adhesion of SE in pretreated human HT29 and Caco-2 cells, porcine IPEC-J2 cells, and chicken CPCE cells. The effect differed from the results of treatment with individual strains of lactobacilli. The reliability of the differences in SE adhesion to intact human and animal enterocytes and to LF3872-and-LS7247-consortium-pretreated enterocytes was significant (*p* < 0.001).

### 2.2. Inhibition of SE Invasion into Intact Human and Animal Enterocytes by LF3872 and LS7247 Strains and Their Consortium

SE possessed high invasive properties into intact human HT29 cells (4.5 ± 0.1 log CFU/mL), intact human Caco-2 cells (4.7 ± 0.1 log CFU/mL), intact porcine IPEC-J2 cells (4.4 ± 0.1 log CFU/mL), and intact chicken CPCE cells (4.3 ± 0.1 log CFU/mL) (Figure 2). The invasion of SE was significantly reduced (more than 2 log CFU/mL) when human and animal enterocytes were LF3872-strain-pretreated or LS7247-strain-pretreated (*p* < 0.01). The pretreatment of human HT29 and Caco-2, porcine IPEC-J2, and chicken CPCE cells with the LF3872 and LS7247 consortium more effectively inhibited the invasion of SE compared to individual strains of lactobacilli that were part of the consortium *(p* < 0.001).

### 2.3. Inhibition of SE Adhesion to Activated Human and Animal Enterocytes by Consortium of LF3872 and LS7247 Strain

The pretreatment of intact human and animal enterocytes with *Salmonella* lipopolysaccharides (LPSs) stimulated SE adhesion to the surface of activated cells. The SE pathogen showed a significant increase in adhesive properties to human HT-29 cells activated by *Salmonella* LPS (the adhesion level was 8.4 ± 0.1 log CFU/mL), to activated human Caco-2 cells (the adhesion level was 8.2 ± 0.1 log CFU/mL), to activated porcine IPEC-J2 cells (the adhesion level was 7.1 ± 0.1 log CFU/mL), and to activated chicken CPCE cells (the adhesion level was 7.4 ± 0.1 log CFU/mL) (Figure 3). The reliability of the SE adhesion growth to activated enterocytes compared with intact human and animal enterocytes was *p* < 0.05. The pretreatment of human HT29 and Caco-2, porcine IPEC-J2, and chicken CPCE cells with the consortium of LF3872 and LS7247 strains led to the cancellation of the ability of *Salmonella* LPS to stimulate the SE adhesion to human and animal enterocytes (Figure 3).

### 2.4. Inhibition of SE Invasion into Activated Human and Animal Enterocytes by Consortium of LF3872 and LS7247 Strains

The pretreatment of intact human and animal enterocytes with *Salmonella* LPS stimulated the SE invasion into activated cells. The SE pathogen showed a significant increase of invasion into activated human HT-29 cells (the invasion level was 6.1 ± 0.1 log CFU/mL), activated human Caco-2 cells (the invasion level was 6.5 ± 0.1 log CFU/mL), activated porcine IPEC-J2 cells (the invasion level was 5.8 ± 0.1 log CFU/mL), and activated chicken CPCE cells (the adhesion level was 6.3 ± 0.1 log CFU/mL) (Figure 4). The SE significantly increased the invasion into activated enterocytes compared with intact human and animal enterocytes (*p* < 0.05). The pretreatment of human HT29 and Caco-2, porcine IPEC-J2, and chicken enterocytes with the consortium of LF3872 and LS7247 strains resulted in the inability of *Salmonella* LPS to stimulate the SE invasion into human and animal enterocytes.

### 2.5. Anti-Salmonella Activity of LF3872 and LS7247 Consortium in Co-Culture Test

The results of the co-cultivation of SE with a consortium of LF3872 and LS7247 strains and individual strains that are part of it are shown in Figure 5. When co-culturing SE with a consortium for 4 h, the number of living pathogen cells decreased from 3.0 log CFU/mL to 2.4 ± 0.2 log CFU/mL. The total number of live cells of the lactobacillus consortium (LF3872 + LS7247) during the 4 h co-cultivation with SE increased from 5.4 log CFU/mL to 7.2 ± 0.2 log CFU/mL. At 6 h of co-culture, the number of live cells of the lactobacillus consortium (LF3872 + LS7247) during co-cultivation with SE increased to 7.9 ± 0.2 log CFU/mL. The number of living SE in the co-culture with a consortium sharply decreased to an undetectable level (<10 CFU/mL) by the end of the experiment. The number of living SE in the mono-culture increased to 6.8 ± 0.4 log CFU/mL. After 6 h of co-culture, the number of live cells of the lactobacillus consortium (LF3872 + LS7247) slowly increased and reached the maximum total cell count at 24 h of the co-culture. When co-culturing SE with the LF3872 strain for 8 h, the number of living SE sharply decreased to an undetectable level (<10 CFU/mL) by the end of the experiment. The number of living SE in the mono-culture increased to 7.3 ± 0.4 log CFU/mL. When co-culturing SE with the LS7247 strain for 10 h, the number of living SE sharply decreased to an undetectable level (<10 CFU/mL) by the end of the experiment. The number of living SE in the mono-culture increased to 7.9 ± 0.4 log CFU/mL.

The consortium of strains had a more pronounced anti-*Salmonella* activity compared to the strains included in its composition.

### 2.6. The Effect of Cell-Free Supernatant from LF3872 and LS7247 Consortium on the Viability of Human HT-29 and Caco-2, Porcine IPEC-J2, and Chicken CPCE Cells

In these experiments, a lyophilized CFS was used, due to the fact that, in the lyophilized state, it retains its biological properties for a long time. The influence of various concentrations of the cell-free supernatant (CFS) (2 µg/mL, 4 µg/mL, 6 µg/mL) from the LF3872 and LS7247 consortium on the viability of human HT-29 and Caco-2, porcine IPEC-J2, and chicken CPCE cells has been studied. In preliminary experiments, it was found that the CFS, at a concentration of 6 µg/mL, had a bactericidal effect on the SE pathogen (*S.* enteritidis Egg 6235 strain) used in this work. In subsequent experiments, we used the CFS additionally in two sub-inhibitory concentrations of 2 µg/mL and 4 µg/mL. The CFS in the concentrations used (2 µg/mL, 4 µg/mL, 6 µg/mL) did not reduce the viability of human HT-29 and Caco-2, porcine IPEC-J2, and chicken CPCE cells (Figure 6).

### 2.7. The Effect of CFS from LF3872 and LS7247 Consortium on the Expression of SE Genes Responsible for Intestinal Colonization and Virulence

The effect of a sub-inhibitory concentration (SIC) (2 and 4 μg/mL) of the CFS from the LF3872 and LS7247 consortium on the expression of SE genes responsible for intestinal colonization and virulence is shown in Table 1. The CFS downregulated the expression of SE genes that are critical for virulence and the colonization of human HT-29 and Caco-2, porcine IPEC-J2, and chicken CPCE cells. The expression of SE genes responsible for the adherence and invasion (*sopB* and *invH*), functions of a Type 3 secretion system (*sipB*, *pipB*, *orf245*, *sipA*, and *ssaV*), cell wall and cell membrane integrity (*ompR*), exo/endonuclease activity (*xthA*), and LPS biosynthesis (*rfbH*) was significantly downregulated (*p* < 0.001).

### 2.8. Consortium of LF3872 and LS7247 Strains Suppressed SE-Induced Production of IL-8, TNF-α, and Il-1β in Human HT-29 and Caco-2, Porcine IPEC-J2, and Chicken CPCE Cells

The inflammatory status of human HT29 and Caco-2, porcine IPEC-J2, and chicken CPCE cells was evaluated by measuring the production levels of pro-inflammatory cytokines (IL-8, TNF-α, and Il-1β) in the culture medium. SE treatment of intact human HT29 and Caco-2, porcine IPEC-J2, and chicken CPCE cells for 24 h resulted in the increased levels of IL-8 (*p* < 0.001), TNF-α (*p* < 0.001), and Il-1β (*p* < 0.001) (Figure 7). The preliminary co-cultivation of human HT29 and Caco-2, porcine IPEC-J2, and chicken CPCE cells with the LF3872 and LS7247 consortium for 6 h, followed by the infection of enterocytes with the SE pathogen for 24 h, led to the complete cancellation of the increase in the level of pro-inflammatory cytokines in the CFS. LF3872-and-LS7247-consortium-pretreated human HT29 and Caco-2, porcine IPEC-J2, and chicken CPCE cells did not react by increasing the production of pro-inflammatory cytokines during their SE infection, unlike intact enterocytes. In the consortium-pretreated and then SE-infected enterocytes, the concentration of pro-inflammatory cytokines in the CFS was significantly lower compared to intact SE-infected enterocytes: IL-8 (*p* < 0.001), TNF-α (*p* < 0.001), and IL-1β (*p* < 0.001).

### 2.9. Consortium of LF3872 and LS7247 Strains Suppressed SE-Induced Expression of TLR4 and Stimulated TLR9 Expression in Human HT-29 and Caco-2 and Porcine IPEC-J2 Cells, and TLR21 in Chicken CPCE Cells

TLR4 mRNA expression was investigated in human HT-29 and Caco-2, porcine IPEC-J2, and chicken CPCE cells (Figure 8). Enterocytes were treated with the LF3872 and LS7247 consortium (2 × 10^8^ CFU/mL, containing 1 × 10^8^ CFU/mL LF3872 strain + 1 × 10^8^ CFU/mL LS7247 strain) or individual strains of lactobacilli: LF3872 (2 × 10^8^ CFU/mL) and LS7247 (2 × 10^8^ CFU/mL). After a 2 h incubation of enterocytes with individual strains of lactobacilli or with their consortium, SE was added to enterocytes. After a 12 h incubation of the mixture with SE, a quantitative real-time polymerase chain reaction (qRT-PCR) was performed to measure the TLR4 mRNA expression levels in human HT-29 and Caco-2, porcine IPEC-J,2 and chicken CPCE cells. After a 24 h incubation of the mixture with SE, qRT-PCR was performed to measure the TLR9 mRNA expression levels in human HT-29 and Caco-2, and porcine IPEC-J2 enterocytes, and TLR21 mRNA expression levels in chicken CPCE cells. In chickens, TLR21 performs functions similar to those of TLR9 in mammals [60]. Untreated HT-29, Caco-2, IPEC-J2, and CPCE cells were used as a control. The SE pathogen stimulates TLR4 mRNA expression in human HT-29 and Caco-2, porcine IPEC-J2, and chicken CPCE cells.

It was found that the LF3872 and LS7247 strains reduce the development of SE-activated TLR4 mRNA expression in human HT-29 and Caco-2, porcine IPEC-J2, and chicken CPCE cells (*p* < 0.05). The consortium of LF3872 and LS7247 strains effectively prevents the development of SE-activated TLR4 mRNA expression in human HT-29 and Caco-2, porcine IPEC-J2, and chicken CPCE cells (*p* > 0.001). TLR9 mRNA expression was investigated in human HT-29 and Caco-2, and porcine IPEC-J2 enterocytes, and TLR21 mRNA expression was investigated in chicken CPCE cells (Figure 9). The SE pathogen does not have the ability to stimulate TLR9 mRNA expression in intact human HT-29 and Caco-2, and porcine IPEC-J2 enterocytes, and TLR21 mRNA expression in intact chicken CPCE cells. However, the addition of the SE pathogen to human and animal enterocytes shielded by individual LF3872 and LS7247 strains of lactobacilli or their consortium stimulated the growth of TLR9 mRNA expression compared to the control (Figure 7).

For human HT-29 and Caco-2, and porcine IPEC-J2 enterocytes, the reliability of differences in TLR9 mRNA growth in the control and in the experiment with individual LF3872 and LS7247 strains was *p* < 0.01. For chicken CPCE cells, the reliability of the differences in TLR21 mRNA growth in the control and in the experiment with individual LF3872 and LS7247 strains was *p* < 0.05. For human HT-29 and Caco-2, and porcine IPEC-J2 enterocytes, the reliability of the differences in TLR9 mRNA growth in the control and in the experiment with the consortium of LF3872 and LS7247 strains was *p* < 0.001. For chicken CPCE cells, the reliability of the differences in TLR21 mRNA growth in the control and in the experiment with the consortium of LF3872 and LS7247 strains was *p* < 0.01. The consortium of LF3872 and LS7247 strains has a more pronounced activity compared to the individual strains that are part of it. The results obtained indicate the ability of the LF3872 and LS7247 consortium to control the intestinal innate immunity using the mechanism of reciprocal regulation of TLR4/TLR9 expression in human and porcine enterocytes and TLR4/TLR21 expression in chicken enterocytes.

### 2.10. The LF3872 and LS7247 Consortium Protected the Intestinal Barrier and Reduced LPS-Induced Paracellular Permeability in Human HT-29 and Caco-2, Porcine IPEC-J2, and Chicken CPCE Cells

The LF3872 and LS7247 consortium inhibited the process of the LPS-induced reduction of the transepithelial electrical resistance (TEER) of the epithelial monolayers formed by human HT-29 and Caco-2, porcine IPEC-J2, and chicken CPCE cells (Figure 10). *Salmonella* LPS reduced the TEER of the intact human HT-29 monolayer from 250 ± 8 to 92 ± 6 ohms (*p* < 0.01). The consortium added to the culture medium of the HT-29 monolayer before the introduction of LPS effectively prevented the LPS-induced TEER reduction process. The TEER differences between the control group of the HT-29 monolayer and the experimental group (LPS + C) were unreliable (*p* > 0.05). *Salmonella* LPS reduced the TEER of the intact human Caco-2 monolayer from 290 ± 8 to 135 ± 7 ohms (*p* < 0.01). The consortium added to the culture medium of the Caco-2 monolayer before the introduction of LPS effectively prevented the LPS-induced TEER reduction process. The TEER differences between the control group of the Caco-2 monolayer and the experimental group (LPS + C) were unreliable (*p* > 0.05). *Salmonella* LPS reduced the TEER of the intact porcine IPEC-J2 monolayer from 305 ± 9 to 157 ± 6 ohms (*p* < 0.01). The consortium added to the culture medium of the IPEC-J2 monolayer before the introduction of LPS effectively prevented the LPS-induced TEER reduction process. The TEER differences between the control group of the IPEC-J2 monolayer and the experimental group (LPS + C) were unreliable (*p* > 0.05). *Salmonella* LPS reduced the TEER of the intact chicken CPCE monolayer from 145 ± 8 to 43 ± 5 ohms (*p* < 0.01). The consortium added to the culture medium of the chicken CPCE monolayer before the introduction of LPS effectively prevented the LPS-induced TEER reduction process. The TEER differences between the control group of the chicken CPCE monolayer and the experimental group (LPS + C) were unreliable (*p* > 0.05).

The consortium of LF3872 and LS7247 strains inhibited the LPS-induced increase in the paracellular permeability of the epithelial monolayers formed by human HT-29 and Caco-2, porcine IPEC-J2, and chicken CPCE cells (Figure 11). *Salmonella* LPS increased the permeability of the intact epithelial monolayers of human HT-29 and Caco-2, porcine IPEC-J2, and chicken CPCE cells by 34–52% (*p* < 0.01). The consortium added to the culture medium of the epithelial monolayers before the introduction of LPS effectively prevented the LPS-induced increase in paracellular permeability (*p* < 0.01). The differences between the paracellular permeability in the control groups and the experimental groups (LPS + C) were unreliable (*p* > 0.05). SE added to HT-29 and Caco-2 enterocytes stimulated an increase in the zonulin concentration in the culture medium of the epithelial monolayers compared to the control (*p* < 0.01). The pretreatment of the HT-29 and Caco-2 epithelial monolayers with the LF3872 or LS7247 strains, which are part of the consortium, reduced the SE-stimulated growth of zonulin secretion (*p* < 0.05). The pretreatment of the HT-29 and Caco-2 epithelial monolayers by a consortium canceled the SE-stimulated growth of zonulin secretion in the culture medium (*p* < 0.01) (Figure 12).

### 2.11. The Effect of CFS from LF3872 and LS7247 Consortium on the Intestinal Alkaline Phosphatase Production and mRNA Expression in Human HT-29 and Caco-2 Enterocytes

The influence of various concentrations of the CFS (2 µg/mL, 4 µg/mL, 6 µg/mL) from the LF3872 and LS7247 consortium on intestinal alkaline phosphatase (IAP) production and mRNA expression in human HT-29 and Caco-2 enterocytes was studied (Figure 13). The preliminary introduction of TNF-α into the culture medium of Caco-2 cells inhibited IAP production and IAP mRNA expression compared to the intact control (*p* < 0.01). The introduction of a 2 µg/mL CFS into the Caco-2 culture medium restored the TNF-α-inhibited IAP production and TNF-α-inhibited IAP mRNA expression to the level of the intact control (*p* < 0.01). The introduction of the CFS at a concentration of 4 µg/mL or 6 µg/mL into the culture medium of Caco-2 cells increased the TNF-α-inhibited IAP production (*p* < 0.001) and TNF-α-inhibited IAP mRNA expression (*p* < 0.01). The preliminary introduction of TNF-α into the culture medium of HT-29 cells inhibited IAP production and IAP mRNA expression compared to the intact control (*p* < 0.001). The introduction of a 2 µg/mL CFS into the HT-29 culture medium restored the TNF-α-inhibited IAP production and TNF-α- inhibited IAP mRNA expression to the level of the intact control (*p* < 0.001). The introduction of the CFS at a concentration of 4 µg/mL or 6 µg/mL into the culture medium of HT-29 cells increased the TNF-α-inhibited IAP production (*p* < 0.001) and TNF-α-inhibited IAP mRNA expression (*p* < 0.001).

## 3. Discussion

SE refers to zoo-anthroponotic nosocomial infections [61], which can colonize processing surfaces and equipment such as stainless steel, marble, and granite, and traditional cleaning and sanitation procedures may not be able to eradicate a pathogen from such surfaces [62,63,64]. Salmonellosis is most often found in piglets weaned from the breast. Infection may be accompanied by the barrier disruption of intestinal epithelial cells caused by *Salmonella* LPS and the development of necrotizing enterocolitis (NEC) with the transition to generalized sepsis (https://iris.who.int/handle/10665/249529, accessed on 20 October 2023). SE strains effectively contaminate eggs and increase chick mortality [65,66,67]. SE would be in a latent stage in the immune-related organs of an animal, such as the spleen and liver. Humans use farm animals in the food chain, and SE can enter the chain and infect humans. When entering maternity hospitals, SE can be an etiological cause of NEC and increase the mortality of newborns [68,69,70]. The traditional treatment of salmonellosis with antibiotics leads to the appearance of MDR strains. The development of innovative consortia based on lactobacilli with bactericidal activity against MDR *Salmonella* strains and the ability to provide intestinal homeostasis are, therefore, very important. The human HT29 and Caco-2, porcine IPEC-J2, and chicken CPCE cells could be considered as in vitro biomodels for the screening of innovative consortia that influence intestinal physiology and play a crucial role in protecting against intestinal infections.

*Salmonella* employs multiple virulence factors to overcome colonization resistance and induce intestinal inflammation [68]. The consortium of LF3872 and LS7247 strains that we created made an important contribution to ensuring intestinal colonization resistance. Two groups can be distinguished in the biological properties of the consortium. The first group of biological properties includes the ability of the consortium to inhibit the adhesion and invasion of the SE pathogen, to exert a bactericidal effect on the SE pathogen, and to inhibit the expression of SE genes responsible for intestinal colonization and virulence. The second group of biological properties of the consortium is aimed at preserving intestinal homeostasis. This group includes the ability of the consortium to inhibit *Salmonella* LPS-induced pro-inflammatory cytokine production and TLR4 mRNA expression, stimulate TLR9 expression, protect the intestinal barrier, and stimulate IAP production by enterocytes.

SE, as an intracellular pathogen, provokes an excessive immune response of the host by activating innate immunity to elude immune control. LPS secreted by *Salmonella* is an agonist of TLR4 receptors expressed on enterocytes [71]. LPS induces a strong pro-inflammatory response from the innate immune system. The interaction of the agonist with TLR4 stimulates the expression of mannose receptors on the surface of enterocytes [72]. The monosaccharide mannose acts as a ligand for the FimH domain found in the type I fimbria of *Salmonella*. The FimH domain is responsible for recognizing mannose patterns on the surface of host enterocytes and facilitating the mannose-dependent adhesion of foodborne pathogens, including *Salmonella* [73,74,75,76,77,78]. Several studies indicate that lactic acid bacteria (LAB) could prevent the attachment of pathogens, in this way reducing colonization, and prevent infection [73,79,80,81]. We found that the LF3872 and LS7247 consortium effectively inhibited the adhesion of the MDR SE strain to intact and activated by *Salmonella* LPS human HT29 and Caco-2, porcine IPEC-J2, and chicken CPCE cells (Figure 1 and Figure 3). The inhibition of adhesion led to a significant decrease in SE invasion into human and animal enterocytes (Figure 2 and Figure 4).

The pretreatment of enterocytes by a consortium reduced the adhesion of SE pathogen by 5–6 log, and invasion decreased by 4 log compared to intact enterocytes. The adhesion of SE to enterocytes leads to Type III secretory system (T3SS) activation. T3SS is a syringe-like apparatus with a needle that ensures the invasion of enterocytes. SE uses T3SS to translocate bacterial proteins into host cells. Structural proteins form the molecular syringe structure of T3SS. The SE pathogen injects effector proteins SipA, SopA, SopB, SopD, and SopE2 through a needle into the host cell, where they trigger cytoskeletal rearrangement and bacterial uptake [82,83]. T3SS effectors stimulate inflammation [84]. During the invasion process, the transmission of signals through the pathogen-associated molecular pattern LPS (agonist of TLR4) causes inflammation. Thus, SE, when ingested into the intestine, induces an inflammatory cytokine storm [85] and reduces the amount of resident microbiota [86,87,88,89], thereby making already existing resources available for their nutritional needs.

The LF3872 genome and LS7247 genome contain genes responsible for the production of factors that increase the SE OM permeability (lactic acid, bacteriocin of class IIb salivaricin, and bacteriocin nizin), as well as genes of peptidoglycan-degrading enzymes enterolysin A and metalloendopeptidase [56,58]. An increase in the OM permeability opens the access of enterolysin A and metalloendopeptidase to the SE cell wall peptidoglycan. The consortium showed pronounced bactericidal activity against the SE pathogen in co-culture tests. The number of living SE in the co-culture with the consortium sharply decreased to an undetectable level (<10 CFU/mL) by the end of the experiment (Figure 5). The number of living SE in the mono-culture increased to 6.8 ± 0.4 log CFU/mL. In other studies, the similar inhibition of SE pathogen growth in a co-culture with a probiotic was noted no earlier than after 8 h of co-cultivation [90].

We found that the CFS from the LF3872 and LS7247 consortium downregulated the expression of SE genes that are critical for virulence and the colonization of human HT-29 and Caco-2, porcine IPEC-J2, and chicken CPCE cells. SE genes responsible for adherence and invasion (*sopB*, *invH*), the functions of T3SS (*sipB*, *pipB*, *orf245*, *sipA*, and *ssaV*), and LPS biosynthesis (*rfbH*) were significantly downregulated by the CFS (*p* < 0.001) (Table 1). In addition to bactericidal anti-*Salmonella* activity, the consortium had the ability to restore and preserve intestinal homeostasis. The intestinal inflammatory cytokine storm caused by SE allows this pathogen to compete with the commensal microbiota, reduce its amount, and effectively colonize the intestine [68,85,91,92].

The consortium inhibited the production of pro-inflammatory cytokines IL-8, TNF-α, and IL-1β, and TLR4 mRNA expression in human and animal enterocytes (Figure 7 and Figure 8). It stimulated the expression of TLR9 in human and porcine enterocytes and stimulated the expression of TLR21 (analog of TLR9) in chicken enterocytes (Figure 9). There are reciprocal relationships between TLR4 and TLR9. An increase in the production of TLR9 inhibits the production of TLR4 [93]. Moshiri M. and colleagues found that the *L. acidophilus* PTCC 1643 strain was able to suppress the inflammation caused by SE infection in HT29 cells and reduce TLR4 mRNA expression [94]. However, the expression of TLR9 has not been studied. TLR9 is important for the protection and repair of intestinal epithelium and has an anti-inflammatory effect [95], and is also involved in intestinal homeostasis [96].

The consortium protected the intestinal barrier functions through an increase in TEER, kept a tight junction, and reduced the *Salmonella* LPS-induced paracellular permeability in human HT-29 and Caco-2, porcine IPEC-J2, and chicken CPCE cells (Figure 10, Figure 11 and Figure 12). Zonulin is a protein that regulates the permeability of tight junctions between enterocytes [97]. The pretreatment of HT-29 and Caco-2 epithelial monolayers by a consortium canceled the SE-stimulated growth of zonulin secretion in the culture medium (*p* < 0.01) (Figure 12). The barrier of the intestinal mucosa is an integral part of maintaining the homeostasis of the intestinal microenvironment [98]. It was previously shown that a multi-species probiotic strain mixture (seven live probiotic species) enhanced the intestinal barrier function by regulating the inflammation and tight junction in LPS-stimulated Caco-2 cells [99]. An investigation of the multi-species probiotic strain mixture on porcine and chicken enterocytes has not been carried out yet.

We found that the CFS of the consortium stimulates the production of the intestinal homeostatic factor IAP in Caco-2 and HT-29 enterocytes (Figure 13). TNF-α inhibited IAP production (Figure 13). Malo M. and colleagues found that TNF-α inhibited the expression of the *iap* gene [100]. The pretreatment of Caco-2 and HT-29 enterocytes with the CFS of the consortium canceled the inhibitory effect of TNF-α on IAP production in intestinal cells. IAP is a leading player in the maintenance of gut homeostasis, including the growth of intestinal microbiota and intestinal barrier function through its ability to dephosphorylate LPS [101,102,103,104,105]. Dephosphorylation leads to the loss of LPS’s ability to bind to TLR4, to be its agonist, and to over-activate innate immunity. Due to the dephosphorylating activity of IAP, it should be included in the molecular factors of innate immunity.

Thus, the consortium exerts a bactericidal effect on the MDR SE pathogen and has the ability to maintain intestinal homeostasis.

## 4. Materials and Methods

### 4.1. Bacterial Strains and Growth Conditions

A list of bacteria used in this work and their growth conditions are provided in Table 2.

### 4.2. Intestinal Epithelial Cells and Growth Conditions

#### 4.2.1. Human Intestinal Epithelial Cells

Immortalized human Caco-2 small intestine epithelial cell line, a relevant in vitro model system for intestinal pathogen–host cell interactions [106], was used. An immortalized human HT-29 large intestine epithelial cell line, a relevant in vitro model system for intestinal pathogen–host cell interactions [107,108,109], was used.

A culture medium consisting of DMEM plus 10% fetal calf serum (FCS) and 0.02% penicillin and streptomycin each was used for immortalized Caco-2 and HT-29 human intestinal epithelial cells. The cells were then seeded into 12-well cell culture plates at a density of 5 × 10^5^ cells/mL to establish a cell monolayer. The plates were incubated for 48 h at 37 °C under 5% CO_2_. To study the monolayer of Caco-2 and HT-29 cells as an intestinal barrier, each cell line was cultured for 15 days with daily replacement of the culture medium.

#### 4.2.2. Porcine Intestinal Epithelial Cells

Immortalized epithelial cells from the porcine intestine, IPEC-J2, were employed as a relevant in vitro model system for porcine intestinal pathogen–host cell interactions [110]. IPEC-J2 cells were suspended and seeded following a similar procedure described in Section 4.2.1. To study the monolayer of IPEC-J2 as an intestinal barrier, cells were cultured for 15 days with daily replacement of the culture medium.

#### 4.2.3. Chicken Primary Cecal Enterocytes

Chicken primary cecal enterocytes (CPCE) is a relevant in vitro model system for chicken intestinal pathogen–host cell interactions [111].

Epithelial cells were isolated from the cecal of 2-week-old chicks (Cobb-500 cross) using a well-established protocol [112]. The cells were then suspended in a culture medium consisting of DMEM, 2.5% FCS, 0.1% insulin, 0.5% transferrin, 0.007% hydrocortisone, 0.1% fibronectin, and 0.02% penicillin and streptomycin each. Primary epithelial cells were suspended and seeded according to the same procedure described in Section 4.2.1. To study the monolayer of CPCE as an intestinal barrier, cells were cultured for 8 days with daily replacement of the culture medium.

### 4.3. Adhesion Assay

Immortalized human HT29 and Caco-2 intestinal epithelial cells, immortalized porcine IPEC-J2 cells, and chicken CPCE cells 5 × 10^5^ CFU were seeded per well in 24 well plates and incubated overnight at 37 °C with 5% CO_2_. The following day, in experimental group of wells, enterocytes were pretreated by LF3872 or LS7247 strains or their consortium (multiplicity of infection (MOI) of 100:1) and incubated for 2 h at 37 °C with 5% CO_2_, then enterocytes were infected with SE at MOI of 10:1. In control wells, enterocytes were infected with SE at MOI of 10:1. The plates were briefly exposed to centrifugation, and then were incubated for 30 min at 37 °C with 5% CO_2_. Unbound bacteria were aspirated. The wells were exposed to washing six times with phosphate buffered saline (PBS); the cells were lysed with 0.1% Triton X-100 (TX-100). Dilutions of the cell lysates were plated on xylose lysine deoxycholate (XLD) agar plates for determination of CFU/mL of SE. The contact of *Salmonella* with enterocytes for 30 min only provides the process of pathogen adhesion to enterocytes. The process of *Salmonella* invasion into enterocytes does not occur during the first 30 min. After adhesion and colonization of enterocytes, *Salmonella* begins to produce an invasion factor that ensures the invasion of the pathogen into enterocytes [113].

### 4.4. Invasion Assay

Immortalized human HT29 intestinal epithelial cells, immortalized porcine IPEC-J2 intestinal epithelial cells, and chicken CPCE cells 5 × 10^5^ CFU were seeded per well in a 24-well plate and incubated overnight at 37 °C with 5% CO_2_. In experimental group of wells, enterocytes were pretreated by LF3872 or LS7247 strains or their consortium (MOI of 100:1) and incubated for 2 h at 37 °C with 5% CO_2_, then enterocytes were infected with SE at MOI of 10:1.

In control wells, enterocytes were infected with SE at MOI of 10:1. The plates were briefly centrifuged and then incubated for 30 min at 37 °C with 5% CO_2_. After aspirating unbound bacteria, the wells were washed six times with PBS, and the cells were lysed using 0.1% Triton X-100 (TX-100). Dilutions of the cell lysates were plated on xylose lysine deoxycholate (XLD) agar plates to determine the CFU/mL of SE.

### 4.5. Determination of Anti-Salmonella Activity of LF3872 and LS7247 Strains and Their Consortium by Co-Cultivation Method in a Liquid Medium

The antibacterial activity of consortium of LF3872 and LS7247 strains against SE was determined by co-cultivation in TGVC medium (Hi Media, India) at a temperature of 37 ± 1 °C for 24 h in anaerobic conditions [114] with modifications. Briefly, overnight culture of LF3872 (4 × 10^5^ CFU/mL), LS7247 (4 × 10^5^ CFU/mL), and SE (1 × 10^3^ CFU/mL) were inoculated into 10 mL of co-culture TGVC medium. The medium was incubated at 37 °C, without shaking under anaerobic conditions. The co-culture medium containing only LF3872, LS7247, or SE was used as controls. In initial experiments, it was observed that the SE, LF3872, and LS7247 strains exhibited growth on the TGVC medium. The quantification of SE cells cultivated on TGVC medium in monoculture (control) and in the presence of lactobacilli on TGVC medium was monitored at 0, 2, 4, 6, 8, 10, 12, 14, and 24 h through plate counts on XLD agar plates (BD, Franklin Lakes, NJ, USA). All plates were incubated for 24 h at 37 °C under aerobic conditions. At the end of the incubation, the colonies of all SE were counted and expressed as colony-forming units per milliliter (CFU/mL).

### 4.6. Preparation of CFS from Consortium of LF3872 and LS7247 Strains

CFS was prepared as previously described [115] in modification. Briefly, LF3872 and LS7247 strains were grown separately overnight in Man–Rogosa–Sharp (MRS) broth under anaerobic conditions at 37 °C. Overnight culture of each strain was used to obtain a consortium containing 1 × 10^8^ CFU/mL of LF3872 + 1 × 10^8^ CFU/mL of LS7247 in the MRS broth and then grown anaerobically for 48 h. CFS was obtained using centrifugation at 5000× *g* for 20 min at 4 °C, filter-sterilized using a 0.22 µm pore size filter (Millipore, Billerica, MA, USA), and concentrated using a Rotational Vacuum Concentrator RVC2-18 (Christ, Osterode am Harz, Germany) through speed-vacuum drying. Lyophilized sediment of CFS from consortium of LF3872 and LS7247 strains was used in the experiments.

### 4.7. Determination of SIC of CFS of LF3872 and LS7247 Consortium

SIC of the lyophilized CFS against SE was determined according to [116,117] with modifications. A 96-well Polystyrene 96-well tissue culture plate (Corning, NY, USA) containing twofold dilutions of CFS in Tryptic Soy Broth (TSB) (Merck, Darmstadt, Germany) was inoculated with 6.0 log CFU of SE along with a negative control (no CFS), and the plate was incubated at 37 °C for 24 h under aerobic condition. The highest concentration of CFS that did not inhibit SE growth after 24 h of incubation was determined as the SIC_1_. Experiments also used a high concentration of CFS, which was next to SIC_1_ and did not inhibit SE growth after 24 h of incubation (SIC_2_). The growth of SE was determined by measuring absorbance using spectrophotometric microplate reader (Bio-Rad Laboratories, Hercules, CA, USA) at 570 nm.

### 4.8. Effect of CFS on Viability of Human HT29 and Caco-2, Porcine IPEC-J2, and Chicken CPCE Cells

Viability of human HT29 and Caco-2, porcine IPEC-J2, and chicken CPCE cells in the presence of the lyophilized CFS was determined by 3-[4,5-dimethylthiazol-2-yl]-2,5-diphenyltetrazolium bromide (MTT) assay. Enterocytes (10^4^ cells/well) were seeded in a 96-well plate for 48 h at 37 °C in a humidified incubator containing 5% CO_2_ to form a monolayer. The enterocytes were incubated with CFS in the following concentrations: 6 µg/mL (dose had a bactericidal effect on SE pathogen after 24 h of incubation), 4 µg/mL (SIC_1_—highest concentration of CFS that did not inhibit SE growth after 24 h of incubation), and 2 µg/mL (SIC_2_—high concentration of CFS, which was next to SIC_1_ and did not inhibit SE growth after 24 h of incubation) for 4 h at 37 °C. The MTT reagent (10 μL) was added to enterocytes and incubated at 37 °C for 2 h. After removing the supernatant, 100 μL isopropanol (Sigma-Aldrich, Saint Louis, MO, USA) was added and the plate was incubated at room temperature in dark for 1 h. The absorbance at 570 nm was measured spectrophotometrically.

### 4.9. RNA Extraction and cDNA Synthesis

Trizol reagent (Invitrogen, Carlsbad, CA, USA) was used to extract total RNA from Caco-2, HT29, IPEC-J2, and CPCE cells. The RNA purity was assessed spectrophotometrically by measuring absorbance at 260 nm and its ratio relative to that at 280 nm. RNase-free DNase I (Thermo Fisher Scientific, Waltham, MA, USA) was used for total RNA treatments to eliminate DNA contaminants. The integrity of RNA was monitored through electrophoresis on an agarose gel stained with GelRedTM (Biotium, Hayward, CA, USA). Subsequently, cDNA was synthesized from total RNA using the QuantiTect^®^ Reverse Transcription kit (Qiagen, Germantown, MD, USA), according to the manufacturer’s recommendations.

### 4.10. Quantitative Real-Time Polymerase Chain Reaction

QRT-PCR was conducted utilizing SYBR^®^ Premix Ex TaqTM (Takara Biotechnology, Otsu, Japan) on a thermal cycler (StepOnePlusTM; Applied Biosystems, Foster City, CA, USA) through 40 cycles. The reaction mixture (20 µL) comprised 5 µL cDNA, 10 µL Power SYBR^®^ Green PCR master mix (Applied Biosystems, San Francisco, CA, USA), 4 µL RNase-free water, and 0.5 μL each of forward and reverse primers. The PCR protocol involved an initial denaturation cycle at 95 °C for 10 min, followed by 40 cycles at 95 °C for 30 s and 60 °C for 30 s, with a final extension step for 30 s at 72 °C. The results were expressed as mean values averaged from three independent experiments, with the β-actin gene (ACTB) serving as an endogenous control. In all tests, a negative control was implemented to detect contamination. The relative quantity (RQ) of gene expression for the sample was calculated using the 2^−∆∆Ct^ method.

The primer sequences used for this study are shown in Appendix A. ACTB was used as a housekeeping reference gene. Human, porcine, and chicken primers used for qRT-PCR analysis of host immune response are shown in Appendix A.

### 4.11. Analysis of Cytokines in Culture Supernatants of Human, Porcine, and Chicken Enterocytes

Human Caco-2, HT-29, porcine IPEC-J2, and chicken CPCE cells were seeded into 24-well culture plates and used for later experiments until the cell density reached 80% confluence. IL-8, TNF-α, and IL-1β production were measured from the supernatants of enterocytes treated or untreated with SE (1 × 10^3^ CFU/mL) or consortium of LF 38721 (1 × 10^5^ CFU/mL) + LS 7247 (1 × 10^5^ CFU/mL) with SE for 24 h. Production of human and porcine cytokines was measured with an ELISA kits (Thermo Fisher Scientific, Waltham, MA, USA), and chicken cytokines were determined according to the protocol specified for the specific ELISA kits (Genie, Dublin, Ireland). The sensitivity of all ELISA kits was 2–6 pg/mL.

### 4.12. Co-Culture of LF3872, LS7247, Their Consortium, and SE with Human HT29 and Caco-2, Porcine IPEC-J2, and Chicken CPCE Cells

Enterocytes were plated onto 12-well plates at a density of 10^6^ cells per well in Roswell Park Memorial Institute (RPMI) 1640 medium supplemented with penicillin/streptomycin. Following a 24 h incubation period at 37 °C with 5% CO_2_, the culture medium was exchanged with fresh RPMI 1640 medium, this time without penicillin/streptomycin. In experimental group 1, enterocytes were co-incubated with LF3872 and LS7247 at concentration of 2 × 10^8^ CFU/mL each, and their consortium at concentration of (1 × 10^8^ LF3872 CFU/mL + 1 × 10^8^ LS7247 CFU/mL) for 2 h at 37 °C under a 5% CO_2_ atmosphere, and then were infected with SE at a density of 1 × 10^7^ CFU/well and incubated for 2 h at 37 °C under a 5% CO_2_ atmosphere. In experimental group 2, in separate 12-well plates, enterocytes were infected with SE at a density of 1 × 10^7^ CFU/well and incubated for 2 h at 37 °C under a 5% CO_2_ atmosphere. The control group plates incubated with either LF3872, LS7247, or consortium alone were also included in the experiment. After incubation, enterocytes of experimental and control groups were washed thrice with culture media (RPMI 1640 containing penicillin/streptomycin) and each cell pellet was washed twice with sterile PBS (pH 6.7) to quantify the expression levels of TLR4, TLR9, and TLR21.

### 4.13. TEER Measurements

TEER was measured using an epithelial volt-ohm-meter (EVOM WPI, Berlin, Germany) to assess cell monolayer integrity. Measurements were conducted at each instance of culture medium exchange according to the manufacturer’s instructions.

### 4.14. Permeability Studies

To evaluate the paracellular permeability of the intestinal epithelium layer, Fluorescein isothiocyanate–dextran with a molecular weight of 4 kDa (FD4) was employed. Then, 250 µL of FD4 solution (1 mg/mL in Hanks’ Balanced Salt Solution (HBSS)) was added to the apical compartment, and 800 µL of HBSS was added to the basolateral compartment. The basolateral side was transferred to a black 96-well plate (Greiner Bio-One, Frickenhausen, Germany) after a 2 h incubation at 37 °C, 150 µL. HBSS and FD4 solutions served as negative and positive controls, respectively. A plate reader TECAN Infinite M200 (Tecan Trading AG, Männedorf, Switzerland) was used to measure fluorescence intensity at excitation and emission wavelengths of 490 and 520 nm, respectively. The permeability coefficient (*P_app_*) was calculated according to equation:$$P_{app}=\frac{dQ}{dt}\cdot \frac{1}{A}\cdot C_{0}$$
where:
*P_app_* = apparent permeability coefficient [cm/s];*dQ*/*dt* = rate of appearance of FD4 on the basolateral side [µg/s];*A* = surface area of the monolayer [cm^2^];*C*_0_ = initial FD4 concentration in the apical side [µg/mL].


### 4.15. Quantification of Tight Junction Regulator Zonulin

Secreted tight junction regulator Zonulin in the supernatant was quantified by using specific ELISA kits (AssayGenie, Dublin, Ireland) according to the manufacturer’s instructions.

### 4.16. Activity of IAP Measurement

IAP activity was measured as described in [118] in modifications. Caco-2 cells and HT-29 cells were treated with various concentrations of CFS from consortium of LF3872 and LS7247 strains. The cells were washed three times with 0.9% NaCl, and then mixed with cold 50 mM Tris buffer at pH 7.5. Protein collected on ice was homogenized using an injection needle. IAP activity was assessed in the condition of 1.25 mg/mL disodium p-nitrophenol phosphate as a substrate in Tris-HCl buffer at pH 10.0, containing 5 mM MgCl_2_∙6H_2_O and 200 mM Trizma base (Sigma-Aldrich, Saint Louis, MO, USA). The absorbance at 405 nm was measured (after 30 min incubation at 37 °C). IAP activity was calculated as µmol/min using a calibration curve for various concentrations of p-nitrophenol. The value of µmol/min was designated as U and then converted to U/mg protein to normalize IAP activity to the protein concentration. Protein concentration was determined using a kit (BioRad, Hercules, CA, USA).

### 4.17. Statistical Analysis

One-way analysis of variance (ANOVA) was used for analyzing obtained results. The results are presented as mean ± standard deviation (SD) from six independent experiments, each tested in triplicate. Statistical significance was assessed using Student’s *t*-tests, and results were considered significant at *p* < 0.05.

## 5. Conclusions

The consortium of LF3872 and LS7247 strains effectively protected human HT-29 and Caco-2, porcine IPEC-J2, and chicken CPCE cells against MDR SE in several ways, including inhibiting the adhesion of the pathogen and reducing its invasion into enterocytes. The consortium and its CFS had a bactericidal effect on SE, but did not affect the viability of human and animal enterocytes. The CFS inhibited the expression of SE genes critical for intestinal colonization and virulence. The consortium and its CFS made an important contribution to the preservation of intestinal homeostasis. For the first time, we have shown that the CFS stimulated the production of the intestinal homeostatic enzyme IAP in HT-29 and Caco-2 enterocytes, inhibited the production of pro-inflammatory cytokines and the expression of TLR4 in human and animal enterocytes, and also increased the expression of TLR9 in human and porcine enterocytes and the expression of TLR21 in chicken enterocytes. The consortium prevented a *Salmonella* LPS-induced decrease in the TEER and an increase in the paracellular permeability in the monolayers of human and animal enterocytes.

These findings indicate that the consortium of LF3872 and LS7247 strains can be used to develop a strategy for preventing the spread of MDR SE among the population and in farm animals.

## Figures and Tables

**Figure 1 antibiotics-13-00030-f001:**
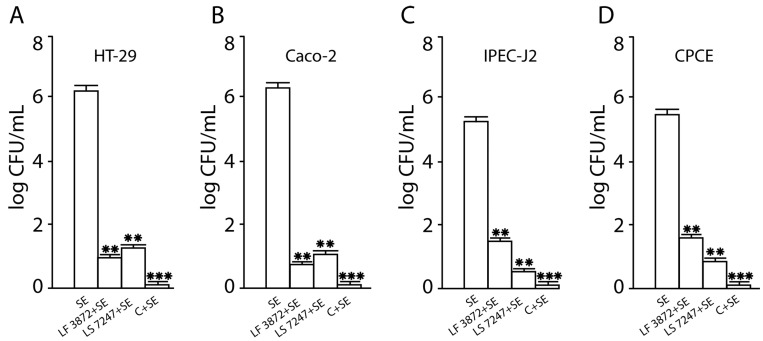
Inhibition of SE adhesion to human HT29 (**A**) and Caco-2 (**B**), porcine IPEC-J2 (**C**), and chicken CPCE (**D**) cells by individual LF3872 strain, LS7247 strain, and their consortium. SE—*S.* enteritidis strain Egg 6235. C—consortium of LF3872 and LS 7247 strains. Data are presented as the means ± SD of six independent experiments, tested in triplicate. ** *p* < 0.01 SE vs. LF3872 + SE, SE vs. LS7247 + SE; *** *p* < 0.001 SE vs. C + SE.

**Figure 2 antibiotics-13-00030-f002:**
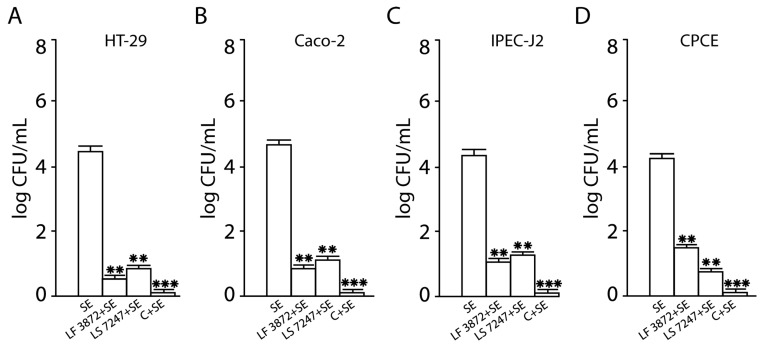
Inhibition of SE invasion into HT29 (**A**), Caco-2 (**B**), IPEC-J2 (**C**), and CPCE (**D**) cells by individual LF3872 strain, LS7247 strain, and their consortium. SE—*S.* enteritidis Egg 6235; C—LF3872 and LS7247 consortium. Data are presented as the means ± SD of six independent experiments, tested in triplicate. ** *p* < 0.01 SE vs. LF3872 + SE, SE vs. LS7247 + SE; *** *p* < 0.001 SE vs. C + SE.

**Figure 3 antibiotics-13-00030-f003:**
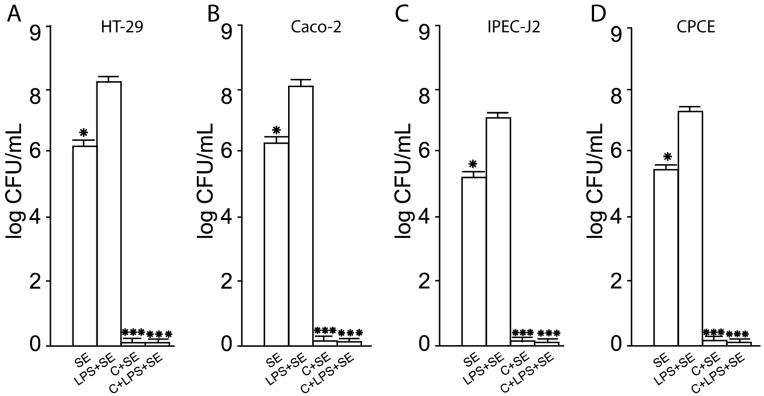
Inhibition of SE adhesion to activated human HT29 (**A**) and Caco-2 (**B**), porcine IPEC-J2 (**C**), and chicken CPCE (**D**) cells by consortium of LF3872 and LS7247 strains. SE—*S.* enteritidis Egg 6235 strain; LPS—lipopolysaccharides of *S.* typhimurium L2262 (Sigma-Aldridge, St. Louis, MO, USA); C—consortium of LF3872 and LS7247 strains. Data are presented as the means ± SD of six independent experiments, tested in triplicate. * *p* < 0.05 SE vs. LPS + SE; *** *p* < 0.001 LPS + SE vs. C + SE, LPS + SE vs. C + LPS + SE.

**Figure 4 antibiotics-13-00030-f004:**
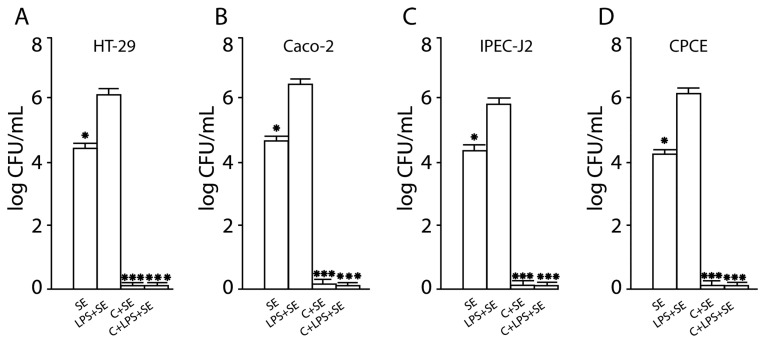
Inhibition of SE invasion into activated human HT29 (**A**) and Caco-2 (**B**), porcine IPEC-J2 (**C**), and chicken CPCE (**D**) cells by consortium of LF3872 and LS7247 strains. SE—*S.* enteritidis Egg 6235 strain; LPS—lipopolysaccharides of *S.* typhimurium L2262 (Sigma-Aldridge, St. Louis, MO, USA); C—consortium of LF3872 and LS7247 strains. Data are presented as the means ± SD of six independent experiments, tested in triplicate. ** p* < 0.05 SE vs. LPS+SE; **** p* < 0.001 LPS + SE vs. C + SE, LPS + SE vs. C + LPS + SE.

**Figure 5 antibiotics-13-00030-f005:**
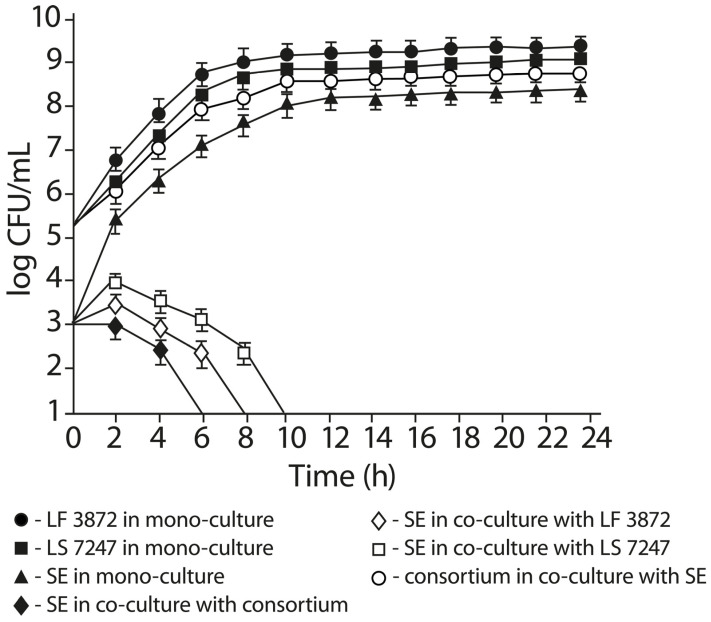
Co-cultivation of LF3872 and LS7247 consortium with SE pathogen. Data are presented as the means ± SD of six independent experiments, tested in triplicate.

**Figure 6 antibiotics-13-00030-f006:**
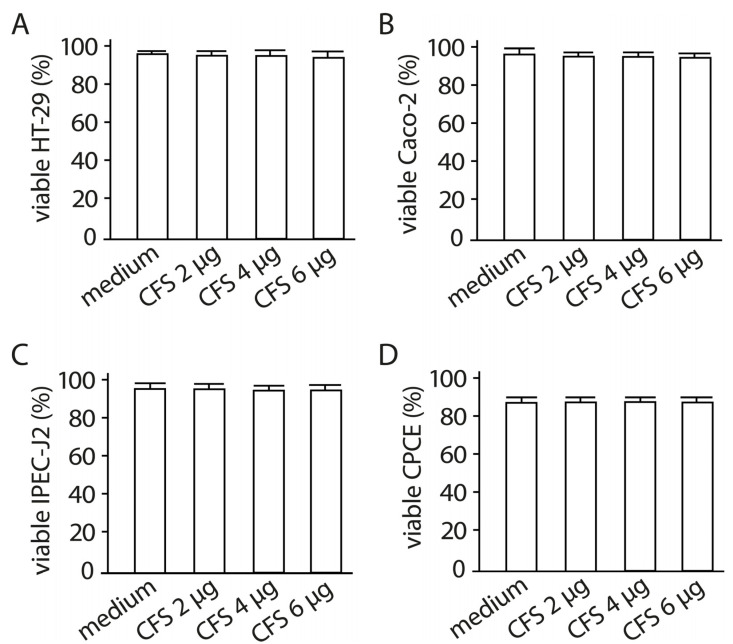
Viability of human HT-29 (**A**) and Caco-2 (**B**), porcine IPEC-J2 (**C**), and chicken CPCE (**D**) cells in the presence of CFS from LF3872 and LS7247 consortium. There are no significant differences in the viability of human HT-29 and Caco-2, porcine IPEC-J2, and chicken CPCE enterocytes in the control and in the experimental groups containing CFS 2 µg and 4 µg. Data are presented as the means ± SD of six independent experiments, tested in triplicate.

**Figure 7 antibiotics-13-00030-f007:**
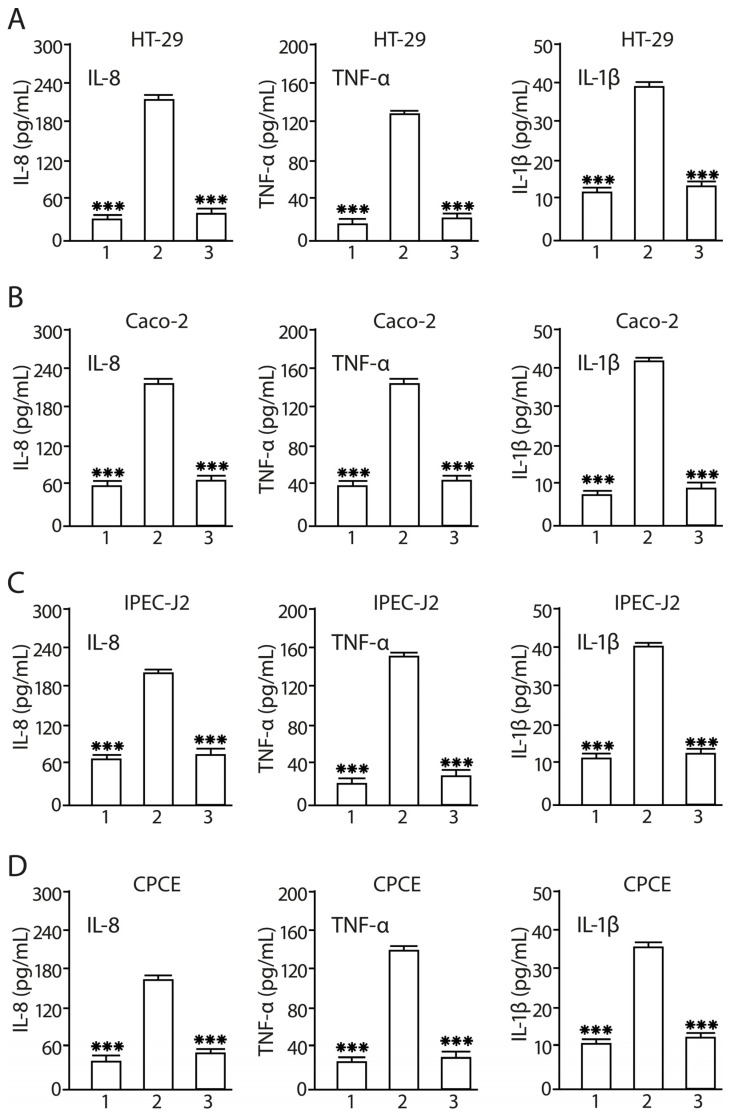
Inhibition of SE-induced production of pro-inflammatory cytokines in human HT-29 (**A**) and Caco-2 (**B**), porcine IPEC-J2 (**C**), and chicken CPCE (**D**) cells by a consortium of LF3872 and LS7247 strains. 1—Control; 2—SE; 3—consortium of LF3872 and LS7247 strains + SE. Data are presented as the means ± SD of six independent experiments, tested in triplicate. *** *p* < 0.001 SE vs. Control; SE vs. C + SE.

**Figure 8 antibiotics-13-00030-f008:**
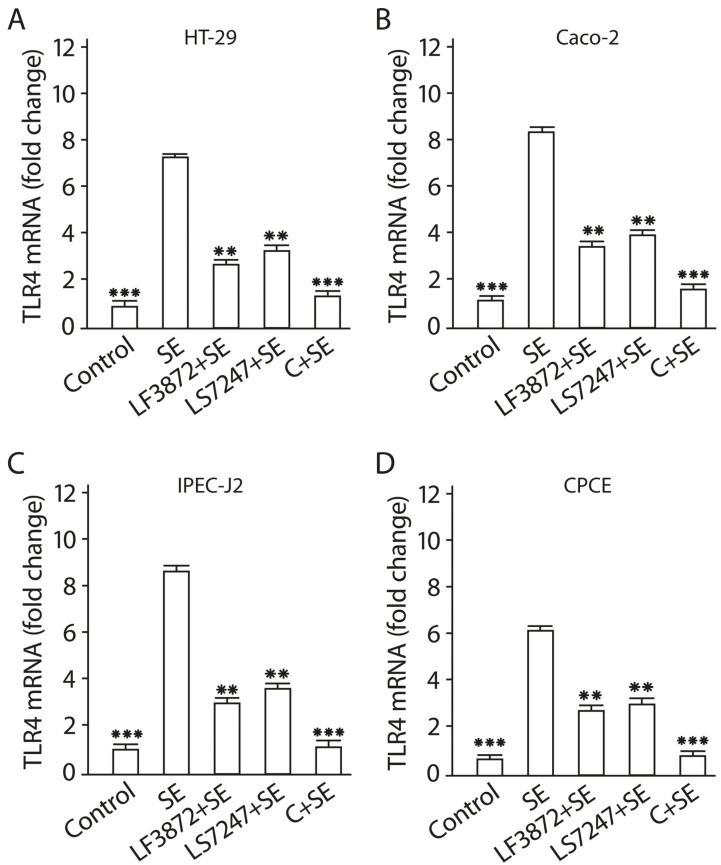
Inhibition of SE-induced TLR4 mRNA expression in human HT-29 (**A**) and Caco-2 (**B**), porcine IPEC-J2 (**C**), and chicken CPCE (**D**) cells by a consortium of LF3872 and LS7247 strains. QRT-PCR was performed to measure TLR4 mRNA expression levels in SE-treated enterocytes after 12 h of incubation. Untreated HT-29, Caco-2, IPEC-J2, and CPCE cells were used as a control. Data are presented as the means ± SD of six independent experiments, tested in triplicate. ** *p* < 0.01 SE vs. LF3872 + SE, SE vs. LS7247 + SE; *** *p* < 0.001 SE vs. control, SE vs. C + SE.

**Figure 9 antibiotics-13-00030-f009:**
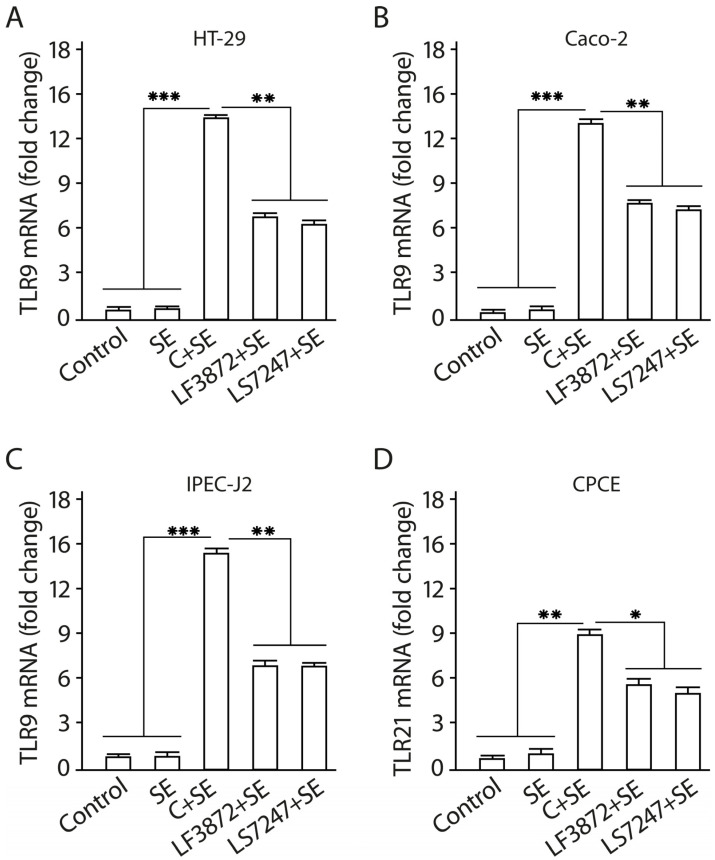
Stimulation of TLR9 expression in human HT-29 (**A**) and Caco-2 (**B**), and porcine IPEC-J2 (**C**) enterocytes, and TLR21 expression in chicken CPCE cells (**D**) by a consortium of LF3872 and LS7247 strains. In SE-treated enterocytes after 24 h of incubation, qRT-PCR was performed to measure TLR9 mRNA expression levels in human and porcine enterocytes and TLR21 expression in CPCE cells. Untreated HT-29, Caco-2, IPEC-J2, and CPCE cells were used as a control. Data are presented as the means ± SD of six independent experiments, tested in triplicate. * *p* < 0.05; ** *p* < 0.01; *** *p* < 0.001.

**Figure 10 antibiotics-13-00030-f010:**
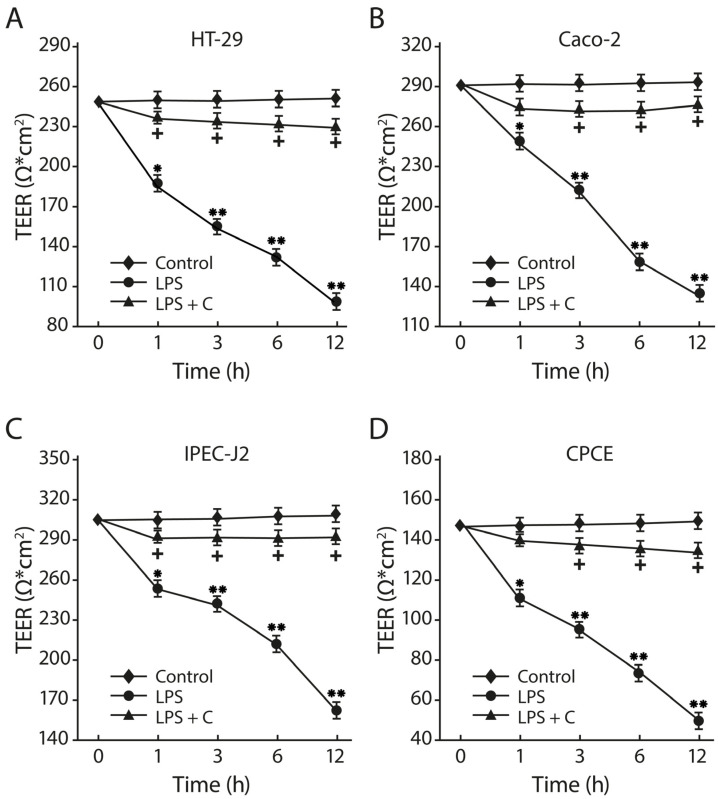
Effects of consortium of LF3872 and LS7247 strains on the intestinal barrier in human HT-29 (**A**) and Caco-2 (**B**), porcine IPEC-J2 (**C**), and chicken CPCE (**D**) enterocytes. C—consortium of LF3872 and LS7247 strains. Data are presented as the means ± SD of six independent experiments, tested in triplicate. * *p* < 0.05 LPS vs. control or LPS + C; ** *p* < 0.01 LPS vs. control or LPS + C; + *p* > 0.05 control vs. LPS + C.

**Figure 11 antibiotics-13-00030-f011:**
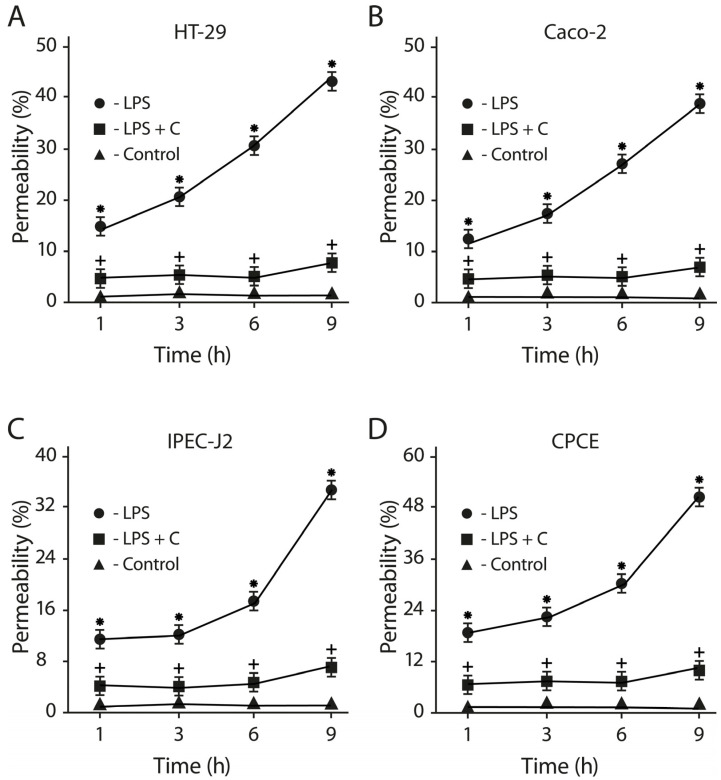
Effects of LF3872 and LS7247 consortium on the intestinal paracellular permeability in human HT-29 (**A**) and Caco-2 (**B**), porcine IPEC-J2 (**C**), and chicken CPCE (**D**) enterocytes. C—consortium of LF3872 and LS7247 strains. Data are presented as the means ± SD of six independent experiments, tested in triplicate. * *p* < 0.05 LPS vs. control or LPS + C; + *p* > 0.05 control vs. LPS + C.

**Figure 12 antibiotics-13-00030-f012:**
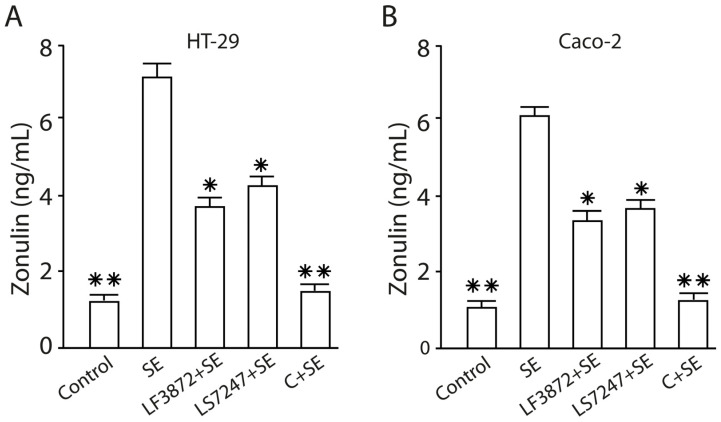
Effects of LF3872 and LS7247 consortium on zonulin, a tight junction protein, secretion by human HT-29 (**A**) and Caco-2 (**B**) enterocytes. Data are presented as the means ± SD of six independent experiments, tested in triplicate. * *p* < 0.05 SE vs. LF3872 + SE or SE vs. LS7247 + SE; ** *p* < 0.01. SE vs. control, SE vs. C + SE.

**Figure 13 antibiotics-13-00030-f013:**
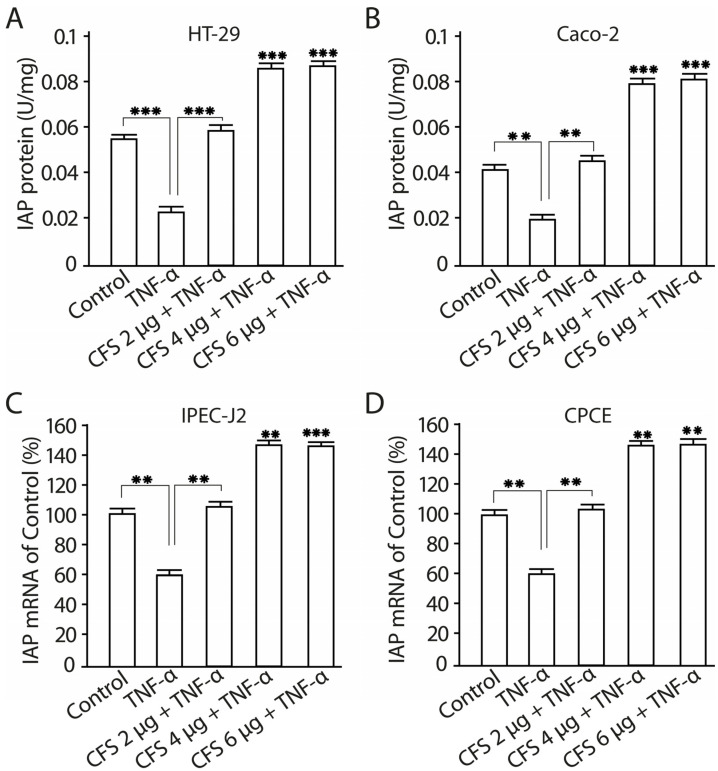
CFS-dependent IAP production in human Caco-2 (**A**) and HT-29 (**B**) enterocytes and IAP mRNA expression in Caco-2 (**C**) and HT-29 (**D**) enterocytes. Data are presented as the means ± SD of six independent experiments, tested in triplicate. ** *p* < 0.01; *** *p* < 0.001.

**Table 1 antibiotics-13-00030-t001:** CFS from LF3872 and LS7247 consortium inhibited expression of SE genes responsible for intestinal colonization and virulence.

Genes	Fold Change *	
	2.0 μg/mL CFS	4.0 μg/mL CFS
*sopB*	−5.7 ± 0.5 ***	−7.2 ± 0.4 ***
*invH*	−3.9 ± 0.2 **	−5.6 ± 0.5 ***
*sipB*	−5.4 ± 0.3 ***	−6.9 ± 0.4 ***
*pipB*	−3.6 ± 0.2 **	−5.2 ± 0.5 ***
*orf245*	−5.6 ± 0.5 ***	−7.8 ± 0.3 ***
*sipA*	−7.9 ± 0.6 ***	−9.5 ± 0.4 ***
*ssaV*	−5.7 ± 0.3 ***	−6.4 ± 0.5 ***
*spvB*	−1.4 ± 0.2 *	−1.8 ± 0.3 *
*mgtC*	−4.2 ± 0.5 **	−7.5 ± 0.4 ***
*sodC*	−3.7 ± 0.4 **	−5.3 ± 0.2 ***
*tatA*	−1.8 ± 0.2 *	−2.6 ± 0.2 *
*hflK*	−1.3 ± 0.2 *	−1.5 ± 0.2 *
*ompR*	−8.4 ± 0.5 ***	−13.7 ± 0.6 ***
*mrr1*	−2.6 ± 0.4 *	−4.8 ± 0.3 **
*lrp*	−1.3 ± 0.2 *	−1.7 ± 0.2 *
*xthA*	−8.4 ± 0.5 ***	−15.6 ± 0.4 ***
*rpoS*	−2.7 ± 0.4 *	−4.2 ± 0.3 **
*ssrA*	−1.9 ± 0.2 *	−2.6 ± 0.2 *
*rfbH*	−9.4 ± 0.6 ***	−11.8 ± 0.5 ***

* The data show fold changes in gene expression with treatments relative to control gene expression. Data are presented as the means ± standard errors of six independent experiments, tested in triplicate. * *p* < 0.05; ** *p* < 0.01; *** *p* < 0.001.

**Table 2 antibiotics-13-00030-t002:** Micro-organisms used in this study.

Micro-organism	Strain	Collection	Antibiotic Resistance	Growth Conditions
*L. fermentum*	LF 3872 ^1^	IIE ^a^		MRS ^b^ 37 °C in CO_2_ incubator at anaerobic atmosphere (5% hydrogen, 10% carbon dioxide, 85% nitrogen) 24–48 h
*L. salivarius*	LS 7247 ^2^	IIE		The same
*S.* Enteritidis	Egg 6235 ^3^	IIE	TET, CHL, LVF, CIP, NAL, SMZ, TMP, AMP	BHI ^c^ 37 °C aerobically 18 h

^1^ Isolate from the milk of a healthy woman. ^2^ Isolate from the intestines and reproductive system of a healthy woman. ^3^ Isolate from chicken’s eggs. ^a^ Collection of Microorganisms at the Institute of Immunological Engineering (IIE), Department of Biochemistry of Immunity and Biodefence, Lyubuchany, Moscow Region, Russia; ^b^ Man–Rogosa–Sharp (MRS) broth or agar-containing plates (HiMedia, India); ^c^ Brain–Heart Infusion (BHI) broth supplemented with 0.5% yeast extract or agar-containing BHI plates. Antibiotic resistance: TET—tetracycline, CHL—chloramphenicol, LVF—levofloxacin, CIP—ciprofloxacin, NAL—nalidixic acid, SMZ—sulfamethoxazole, TMP—trimethoprime, AMP—ampicillin.

## Data Availability

The data are contained within the article and Appendix A.

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
