# Peer review of "Anti-*Salmonella* Defence and Intestinal Homeostatic Maintenance In Vitro of a Consortium Containing *Limosilactobacillus fermentum* 3872 and *Ligilactobacillus salivariu*s 7247 Strains in Human, Porcine, and Chicken Enterocytes"

_antibiotics, 2023, doi:10.3390/antibiotics13010030_

Round 1

Reviewer 1 Report

Comments and Suggestions for Authors

Abramov et al. studied the efficacy of Lactobacillus strains on Salmonella and its LPS induced inflammatory effect in vitro. The Lactobacillus strains were isolated from different source of human samples and showed better activity against Salmonella and its LPS induced inflammatory effect. Authors used appropriate techniques to analyze the anti-inflammatory effect of newly isolated Lactobacillus consortium. No more mistakes found in the manuscript and it may acceptable for the publication.

Reviewer 2 Report

Comments and Suggestions for Authors

In the article "Anti-Salmonella defence and intestinal homeostatic maintenance in vitro of consortium containing Limosilactobacillus fermentum 3872 and Ligilactobacillus salivarius 7247 strains in human, porcine and chicken enterocytes"  by Abramov V.M. et al., the authors report the use of a consortium of strains, L. fermentum 3872 and L. salivarius 7247, to develop a strategy for preventing the spread of MDR-SE in humans and farm animals. The authors state that the consortium, consisting of L. fermentum 3872 and L. salivarius 7247 strains, produces enzymes that destroy peptidoglycans and a complex of factors (lactic acid, nisin, salivaricin) that increase the permeability of the outer membrane of Salmonella, allowing access to peptidoglycan.

For the first time, it was found that the cell-free supernatant (CFS) of the consortium can stimulate the production of the intestinal homeostatic factor - intestinal alkaline phosphatase (IAP) in enterocytes. IAP plays a leading role in maintaining gut homeostasis, influencing the growth of intestinal microbiota, and contributing to intestinal barrier function through its ability to dephosphorylate LPS.

The research results are well presented and analyzed in detail in the section Discussion. The article is qualitatively designed in accordance with the requirements of the journal Antibiotics. I recommend publishing the article in the journal Antibiotics. However, I have a few minor comments and suggestions for improving the text.

Minor comments

1.               In the sentence “The effects of probiotic strains are mediated via immune regulation, especially through modulating the balance between pro- and anti-inflammatory cytokines” indicate in the brackets which cytokines you are referring to.

2.               Add the link to NCB GenBank

3.               Make the letters A, B, etc. in the figures less bold

Reviewer 3 Report

Comments and Suggestions for Authors

Dear Authors,

The original manuscript entitled “Anti-Salmonella Defence and Intestinal Homeostatic Maintenance in vitro of a Consortium Containing Limosilactobacillus fermentum 3872 and Ligilactobacillus salivarius 7247 strains in Human, Porcine, and Chicken Enterocytes” is well-written, structured and developed by Abramov et al. in suitable English with a clear structure. They characterised the genomics analysis and anti-bacterial activities of two lactic acid bacteria isolated from human breast milk and intestine and reproductive systems against pathogenic Salmonella strains. This manuscript is novel and the results are very interesting.  There are some points of view which should be addressed and implemented.

-        The abstract section is very long and it is recommended to summarise this section more and focus more on results and conclusions.

-        Please add the main gap of knowledge (regarding the two stains) before the aim of the study in the last paragraph of the introduction section.

-        In Figure 6: If there is not any significant difference, please indicate it in the figure caption.

-        Please explain in the text your reasons for choosing the specific cell lines: why HT-29? Why caco-2? …

-        Considering the huge data in this study, the discussion section is very short and limited. Please develop this section as well. 

Round 2

Reviewer 3 Report

Comments and Suggestions for Authors

Dear authors,

Thank you for your response and I have no more comments.